# The impact of organic nitrates on summer ozone formation in Shanghai, China

Chunmeng Li[1], Xiaorui Chen[2, 3 *], Haichao Wang[2, 3], Tianyu Zhai[4], Xuefei Ma[5], Xinping Yang[4], Shiyi Chen[5], Min Zhou[6], Shengrong Lou[6], Xin Li[5], Limin Zeng[5], Keding Lu[5*]

[1] Center for Environmental Metrology, The National Institute of Metrology, Beijing 100029, China.
[2] School of Atmospheric Sciences, Sun Yat-sen University, Zhuhai, Guangdong, 519082, China.
[3] Guangdong Provincial Observation and Research Station for Climate Environment and Air Quality Change in the Pearl River Estuary, Key Laboratory of Tropical Atmosphere-Ocean System, Ministry of Education, Southern Marine Science and Engineering Guangdong Laboratory (Zhuhai), Zhuhai, 519082, China.
[4] State Environmental Protection Key Laboratory of Vehicle Emission Control and Simulation, Chinese Research Academy of Environmental Sciences, Beijing, 100012, China
[5] State Key Joint Laboratory of Environmental Simulation and Pollution Control, The State Environmental Protection Key Laboratory of Atmospheric Ozone Pollution Control, College of Environmental Sciences and Engineering, Peking University, Beijing, 100871, China.
[6] State Environmental Protection Key Laboratory of the Cause and Prevention of Urban Air Pollution Complex, Shanghai Academy of Environmental Sciences, Shanghai, 200233, China.

* Correspondence: chenxr95@mail.sysu.edu.cn; k.lu@pku.edu.cn

## Abstract

Organic nitrates serve as important secondary oxidation products in the atmosphere, playing a crucial role in the atmospheric radical cycles and influencing the production of secondary pollutants (ozone ($O_3$) and secondary organic aerosols). However, field measurements of organic nitrates are scarce in China, and a comprehensive localized mechanism for organic nitrates is absent, hindering effective pollution mitigation strategies. In this study, we conducted measurements of ambient gaseous organic nitrates and examined their effects on local $O_3$ production at a polluted urban site in eastern China during summer. The average daytime concentrations of alkyl nitrates (ANs) and peroxy nitrates (PNs) throughout the campaign were 0.5±0.3 ppbv and 0.9±0.7 ppbv, respectively, with peaks reaching up to 1.6 ppbv and 3.6 ppbv. An observation-constrained box model, incorporating an updated mechanism for organic nitrates, was employed to assess the environmental impact of these compounds. The model results indicated that PNs production inhibited the daytime $O_3$ production by 16% (0.8 ppbv/h), which is relatively low compared to previous studies. Furthermore, scenario analyses revealed that production yields (α) of ANs would alter the response of $O_3$ formation to precursors due to varying compositions of volatile organic compounds. Our results suggest that blind pollution control may cause ineffective pollution prevention and highlight the necessity of a thorough understanding on organic nitrate

chemistry for local $O_3$ control strategy.

## 1. Introduction

Tropospheric ozone, as an important oxidant, influences the atmospheric lifetimes of trace gases through its involvement in photochemical processes, thereby playing a crucial role in climate change and atmospheric chemistry. There is a broad consensus that high near-surface ozone concentrations are hazardous to human health and environmental ecosystems, particularly affecting the human respiratory and cardiovascular systems, and result in decreased yields of various crops (Ashmore, 2005; Xue and Zhang, 2023). A scientific assessment of tropospheric ozone is essential for the development of public health policies and for addressing long-term air pollution challenges (Monks et al., 2015). Primary pollutants, such as nitrogen oxides ($NO_x$) and volatile organic compounds (VOCs), participate in the formation of $HO_x$ radicals ($RO_x = RO_2 + HO_2 + OH$) cycles and $NO_x$ cycles under sunlight, leading to the continuous production of ozone as a secondary oxidation product within these cycles. In addition to the reaction between OH and $NO_2$ that produces $HNO_3$ as part of chain termination reactions, the interaction of $RO_2$ and NO that produces organic nitrates is of increasing concern (Present et al., 2020). The atmospheric production of organic nitrates consumes both $NO_x$ and $RO_2$. Therefore, the chemistry of organic nitrates will significantly influence the prevention and control of ozone, with $NO_x$ and VOCs serving as independent variables.

Both anthropogenic activities and natural processes contribute to the emissions of $NO_x$ and VOCs, which produce $RO_2$ in the presence of oxidants such as OH. Subsequently, $RO_2$ reacts with NO to yield $NO_2$ and RO. After that, $NO_2$ photolysis produces $O_3$, while RO is converted into $HO_2$ through an isomerization reaction, thereby forming the ozone production cycle. Within the cycle, a branching reaction between $RO_2$ and NO leads to the formation of alkyl nitrates ($RONO_2$, ANs), while $RO_2$ may also react with $NO_2$ to generate peroxy nitrates ($RO_2NO_2$, PNs). Given that PNs are prone to thermal dissociation near the surface (Roberts and Bertman, 1992), they can influence $O_3$ production by modifying the availability of $NO_x$ and $RO_x$. Due to the competitive production dynamics between PNs and $O_3$, numerous field observations and model simulations have been conducted to investigate the impact of peroxyacetyl nitrate (PAN) on $O_3$ production (Liu et al., 2021; Zeng et al., 2019; Zhang et al., 2020). For ANs formation, the branching ratio ($\alpha$), the reaction ratio $k_{1b}/(k_{1a}+k_{1b})$, varies between 0.1-35%, which are associated with the carbon chain structure of the molecule, the distribution of functional groups, temperature, and pressure (Reisen et al., 2005;Arey et al., 2001;Wennberg et al., 2018;Russell and Allen, 2005;Butkovskaya et al., 2012;Cassanelli et al., 2007). Some values of $\alpha$, which have not been quantified in the laboratory, are estimated through structure-activity relationships (Arey et al., 2001; Reisen et al., 2005; Teng et al., 2015; Yeh and Ziemann, 2014a; Yeh and Ziemann, 2014b). Multiple field observations revealed a strong linear correlation between ANs and $O_3$, with a correlation coefficient ($r^2$) exceeding 0.5, further substantiating the competitive relationship between ANs and $O_3$ (Aruffo et al., 2014; Day et al., 2003; Flocke et al., 1998).

$$RO_2+NO \rightarrow RO+NO_2 \quad (R1a)$$
$$RO_2+NO \rightarrow RONO_2+NO_2 \quad (R1b)$$

Currently, research on the effects of ANs on $O_3$ distribution is predominantly located in Europe and the United States. Following the first in situ measurement of total organic nitrates through thermal dissociation laser-induced fluorescence instrument (TD-LIF) by Day et al., field observations of total ANs have been continuously conducted to study the role of ANs in the nitrogen cycle (Aruffo et al.,

2014; Browne et al., 2013; Chen et al., 2017; Darer et al., 2011; Day et al., 2003; Sadanaga et al., 2016). In conjunction with field observations and model simulations, Farmer et al. were the first to indicate that ANs influence the sensitivity of $NO_x$-VOCs-$O_3$ (Farmer et al., 2011). As $NO_x$ emissions decrease due to pollution control measures, ANs chemistry is expected to play an increasingly significant role in $O_3$ simulations (Present et al., 2020; Zare et al., 2018). Current mechanisms for $O_3$ simulations generally achieve reasonable predictions in large-scale models; however, they exhibit deviations exceeding 10 ppbv in regional simulations (Young et al., 2018). Subsequent studies have demonstrated that refining the ANs chemistry can further improve the simulation performance for $O_3$ (Schwantes et al., 2020). ANs are predominantly produced through oxidation reactions facilitated by OH, $O_3$, and $NO_3$. The daytime ANs are mainly contributed by the OH channel, whereas during nighttime, the contribution of the $NO_3$ channel is linked to significantly increased yields of ANs (Liebmann et al., 2018; Ng et al., 2017; Zare et al., 2018). Presently, the enhancement of ANs chemistry mainly focuses on BVOCs, particularly isoprene and monoterpenes. These researches aim to enhance the yield of ANs derived from BVOCs, the re-release ratio of ANs to $NO_x$, and the contribution of ANs to aerosols (Fisher et al., 2016; Romer et al., 2016; Travis et al., 2016; Zare et al., 2018). Despite the establishment of a complete mechanism scheme at present, significant uncertainties remain in ANs simulation, which may introduce substantial uncertainties into the $O_3$ simulation.

Atmospheric pollution is common across China, particularly in the Yangtze River Delta. Shanghai, as a highly urbanized metropolis in the Yangtze River Delta, has rendered the region's complex pollution due to its rapid economic growth and urbanization (Wang et al., 2022; Zhu et al., 2021). Previous studies have shown a significant increase in near-surface $O_3$ levels from 2006 to 2016 in Shanghai (Gao et al., 2017). However, research on the ANs chemistry and their impact on $O_3$ pollution remains limited in this area. In addition, most field measurements of ANs have focused on short-chain species (Ling et al., 2016; Song et al., 2018; Sun et al., 2018; Wang et al., 2013), which have been observed to exert a typical inhibition effect on daytime $O_3$ production. A limited number of total ANs measurements found that both ANs and $O_3$ production were in the VOC-limited regime (Li et al., 2023). To further investigate the influence of organic nitrates on $O_3$ production, this study describes the distribution of organic nitrates based on a comprehensive field campaign conducted in Shanghai, analyzes the effects of organic nitrates on $O_3$ production through model simulations, and offers recommendations for the prevention and control of ozone pollution in the region.

## 2. Methodology

### 2.1 Measurement site and instrumentations

A comprehensive campaign was conducted in Shanghai to further investigate the chemical behavior of organic nitrates in urban environments across China. As depicted in Fig. 1, the site is located in the Xuhui District of Shanghai (121.44°E, 31.18°N), in proximity to the Shanghai Inner Ring Viaduct, surrounded by numerous residential and office areas without significant industrial emission sources. The site is mainly influenced by morning-evening rush hours, as well as the transport of air masses to the urban location. The overall wind speed was low, predominantly originating from the east. All the measurement instruments were housed in the temperature-controlled room within the laboratory building at the Shanghai Academy of Environmental Sciences. Thermal Dissociation-

Cavity Enhanced Absorption Spectroscopy (TD-CEAS) was positioned on the 7th floor about 25 m above ground level, with the sampling tube extending out through the window.

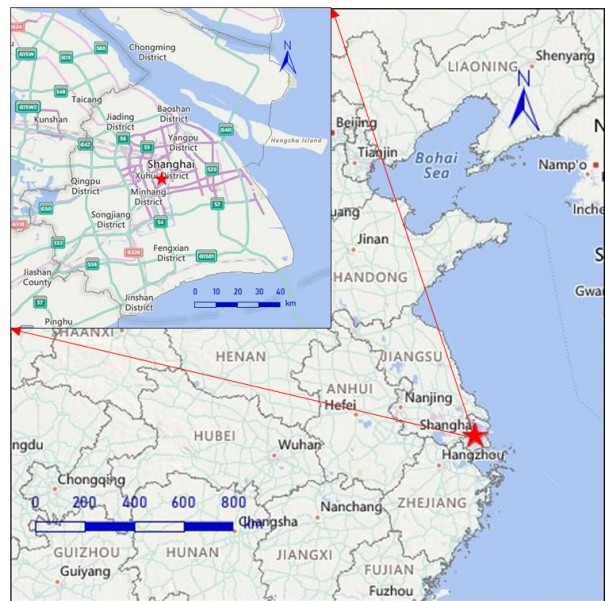

**Figure 1.** Map of the city of Shanghai and the surrounding area (@ MeteoInfoMap). The red star is the location of the campaign site.

The Shanghai campaign focused on studying summer ozone pollution, with the chemical parameters presented in Table 1. Organic nitrates were measured by TD-CEAS with a sampling flow rate of 3 L/min and a sampling duration of 3 min for alternating measurements of $NO_2$, PNs, and ANs. The sampling apparatus consisted of a 2-meter-long 1/4-inch tetrafluoroethylene (TFE) tube, through which the atmosphere was filtered through a TFE particulate filter. The membrane was replaced once a day to mitigate the interference caused by wall loss. The measurement of PAN was conducted by gas chromatography electron capture detection (GC-ECD). The Measurement of $N_2O_5$ was performed via CEAS, which relies on the thermal dissociation of $N_2O_5$ to yield $NO_3$. Particulate nitrates and gaseous $HNO_3$ were measured online by AeRosols and GAses (MARGA), where soluble substances were quantified through ion chromatography following dissolution. The measurements of HONO were finished by CEAS during the campaign. Measurements of VOCs were achieved using a combination of GC-FID and GC-MS, with GC-MS predominating due to the limited species measured by GC-FID. The photolysis rate constant (J value) was determined using a spectrum radiometer with a time resolution of 20 s. Additionally, simultaneous measurements of other trace gases such as NO, $NO_2$, $SO_2$, CO, $O_3$, and $PM_{2.5}$ were conducted using commercial instruments.

**Table 1.** Measured species for organic nitrates analysis and instrument time resolution, accuracy, and detection limitation.

| Parameters | Measurement technique | Time resolution | Accuracy | Detection limit |
|---|---|---|---|---|
| ANs, PNs, $NO_2$ | TD-CEAS | 3 min | ± 8% | 93 pptv |
| PAN | GC-ECD | 5 min | ± 10% | 5 pptv |
| $N_2O_5$ | CEAS | 1 min | ± 19% | 2.7 pptv |
| NO | Thermo 42i | 1 min | ± 10% | 60 pptv |

| NO$_2$ | Chemiluminescence | 1 min | ± 10% | 300 pptv |
|---|---|---|---|---|
| HONO | CEAS | 1 min | ± 3% | 100 pptv |
| Particulate nitrate | 2060 MARGA | 1 h | ± 3% | 0.01 μg/m$^3$ |
| HNO$_3$ | 2060 MARGA | 1 h | ± 3% | 0.01 μg/m$^3$ |
| SO$_2$ | Thermo 43i-TLE | 1 min | ± 16% | 50 pptv |
| HCHO | Hantzsch fluorimetry | 1 min | ± 5% | 25 pptv |
| CO | Thermo 48i-TLE | 1 min | ± 16% | 50 pptv |
| O$_3$ | Thermo 49i | 1 min | ± 5% | 0.5 ppbv |
| PM$_{2.5}$ | Thermo TEOM | 1 min | ± 5% | 0.1 μg/m$^3$ |
| VOCs | GC-FID/GC-MS | 1 h | ± 30% | 20-300 pptv |
| J value | Spectrum radiometer | 20 s | ± 10% | $5 \times 10^{-5}$ s$^{-1}$ |


## 2.2 Model calculation

To investigate the impact of ANs chemistry on O$_3$ production, a box model was employed to
simulate the photochemistry processes. The mechanism of the model was enhanced based on RACM2
(Regional Atmospheric Chemical Mechanism version 2). This box model simulates the
physicochemical processes occurring within a defined volume for each reactant. It utilizes measured
parameters as the boundary condition to simulate the chemistry process while allowing for convenient
adjustments to the mechanism. The model generates files detailing concentration changes, budget
processes, and reaction rates, thereby providing an efficient means to simulate ground-level pollutants.
In this study, the box model was constrained by various parameters, including J values, O$_3$, NO, NO$_2$,
CO, HONO, VOCs, RH, temperature, and pressure, with the time step set to 1h. The deposition process
was quantified using the deposition rate and the boundary layer height, with the dry deposition rate
established at 1.2 cm/s and the boundary layer height constrained by data obtained from NASA.
The RACM2 facilitates classification through the distribution of functional groups and
subsequently delineates reactions involving 17 stable non-organic compounds, 4 inorganic
intermediates, 55 stable organic compounds, and 43 intermediate organic species within the
mechanism. However, the mechanism description for ANs is notably abbreviated. The various ANs,
characterized by differing functional groups, are treated as a unified entity, thereby neglecting the
influence of functional groups on the underlying chemistry. Consequently, this study builds on the
previous research and further evaluates the updates of the mechanism (Li et al., 2023). These
mechanistic updates are developed based on the work of Zare et al. and primarily encompasses the
oxidation processes of BVOCs by OH and NO$_3$, as well as the deposition and the aerosol uptake, which
are detailed in the SI (Zare et al., 2018). Accordingly, three mechanistic schemas are compared based
on the campaign, which will be elaborated upon in subsequent sections. A box model based on the
above mechanism is used to calculate the ozone production rate (P(O$_3$)) (Tan et al., 2017b). P(O$_3$) was
quantified based on the net production rate of O$_x$ (the sum of O$_3$ and NO$_2$), by subtracting the O$_x$
depletion from the instantaneous O$_x$ production. The simulation uncertainty of the box model is about
40%, introduced mainly by the simplified reaction rate constants, photolysis rate constants, and near-
ground deposition (Lu et al., 2013). The impact of PNs photochemistry on local ozone is quantified by
comparing the difference of the daytime P(O$_3$) between the scenarios with and without PNs

photochemistry via a chemical box model. Here, the PNs photochemistry includes the production and removal of PAN, MPAN and PPN.

To facilitate the assessment of the impacts of ANs on local $O_3$ pollution, we further conducted a simplified box model based on the steady-state assumption approach. Several studies have examined the combined effect of $\alpha$ and VOCs reactivity on local $O_3$ levels using this approach (Farmer et al., 2011; Present et al., 2020; Romer et al., 2016; Romer et al., 2018). Briefly, the production pathway of ANs is simplified according to VOCs categories and the production rate of OH and $HO_2$ ($P(HO_x)$) is fixed to a constant value. VOCs are categorized into two primary groups: non-oxygenated VOCs (RVOCs) and oxygenated VOCs (OVOCs). Both categories of VOCs undergo oxidation by OH, resulting in the formation of $RO_2$, specifically $RVOCRO_2$ and $OVOCRO_2$. The interaction between $RVOCRO_2$ and NO will produce $\alpha$ ANs, $(1-\alpha)$ $NO_2$, $HO_2$, and OVOC. Conversely, the reaction of $OVOCRO_2$ with NO directly generates $NO_2$. In the Beijing-Tianjin-Hebei, Yangtze River Delta, and Chengdu-Chongqing regions of China, $P(HO_x)$ is approximately 4 ppbv/h (Lu et al., 2013; Tan et al., 2018a; Tan et al., 2018b). $P(HO_x)$ is therefore assumed to be 4 ppbv/h, with equal production rates of OH and $HO_2$. The model also incorporates additional processes, including inter- and self-reactions of $RO_2$, as well as reactions between $NO_2$ and NO, and deposition processes. In addition, during the daytime, NO is determined by $j(NO_2)$, $O_3$, and $NO_2$ according to the photo-stationary state among $NO$-$NO_2$-$O_3$. Based on the above simplified approach, production rates of ANs and $O_3$ in this study can be derived by direct calculations.

To investigate the effects of $NO_x$ and VOCs on $O_3$ production, the theoretical maximum of $P(O_3)$ was simulated by a box model under varying concentrations of $NO_x$ and VOCs. This approach was employed to develop an empirical kinetic modeling approach for ozone production (EKMA). The EKMA serves as a model sensitivity method to inform strategies for pollutant abatement. In this study, EKMA utilizes the measured mean parameters as the initial point. Each parameter was incrementally adjusted in 30 equidistant steps to create scaled arrays of VOCs and $NO_x$, which were subsequently used to simulate the variations in $P(O_3)$ resulting from changes in precursor concentrations. Ultimately, contour plots illustrating the relationship between $P(O_3)$ arrays versus the concentrations of $NO_x$ and VOCs are plotted based on the simulation results.

## 3. Results and discussions

### 3.1 Overview of organic nitrates and precursors

The duration of the Shanghai campaign was 20 days, spanning from May 25 to June 13, 2021. The analysis of organic nitrates is performed from 6 a.m. to 6 p.m., as measurements taken during nighttime were subject to interference from $N_2O_5$ and its derivatives, a phenomenon noted in previous studies (Li et al., 2021; Li et al., 2023). Simultaneous measurements of PAN and PNs were conducted throughout the campaign. There was a malfunction of the GC-ECD instrument from June 12 to June 13, during which the measurements of PAN were generally low. Relative humidity (RH) varied considerably, with over 95% during rainfall periods on June 2, June 9, June 10, and June 13, while the remaining days were predominantly sunny. Temperatures were high, with minimums of 20 °C and daytime peaks reaching up to 36 °C. The wind speeds were generally high during the daytime and low at night, with maximum of 4.2 m/s. The easterly winds prevailed during the campaign, except for May

210  27-28 and June 3-6 with mostly west and southwest winds.

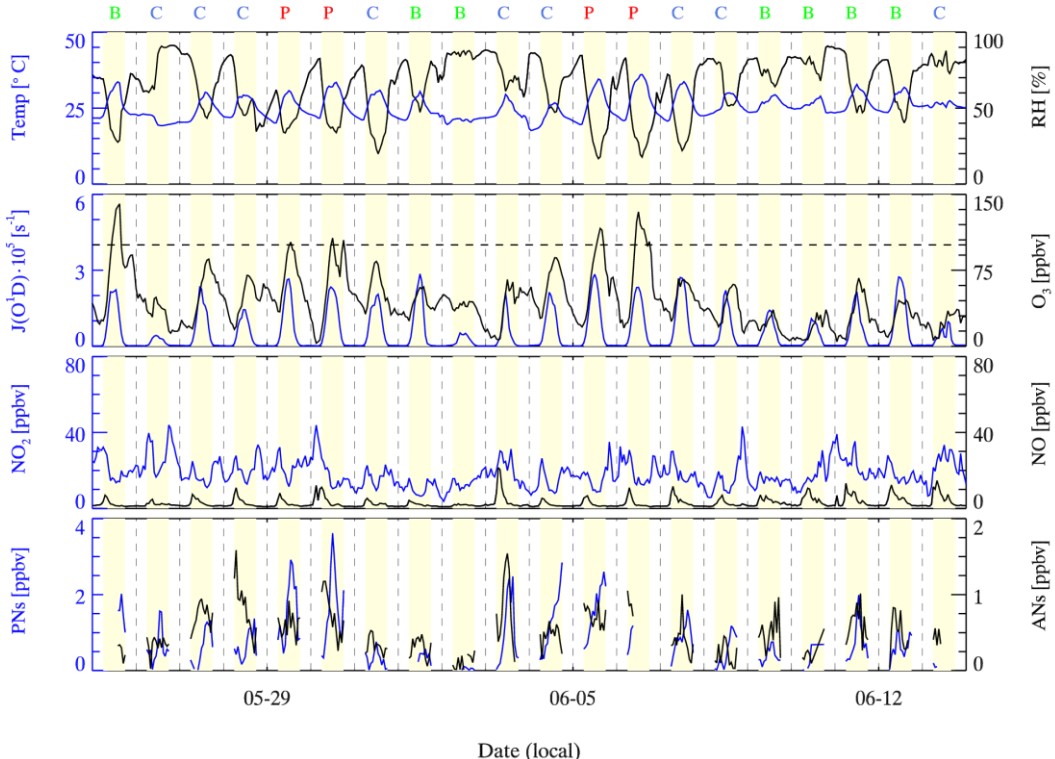

211

**Figure 2.** The time series of the related parameters focused on organic nitrates during the campaign. The background
days are represented by green B, the clean days are represented by blue C, and the ozone pollution day is represented
by red P.

According to Chinese air quality standards for Class II areas, which define ozone pollution days
as those with an hourly average exceeding 100 ppbv, the periods from May 29 to May 30 and June 5
to June 6 have been identified as ozone pollution days. The days without ozone pollution are
categorized as clean or background days. For clean days, parameters, including $K_{OH}$, $SO_2$, and CO,
show significant diurnal variations (Fig S1), and no rain occurs. The days that are neither ozone
pollution days nor clean days are then classified as background days. The daytime averages of
environmental parameters during the ozone pollution period, the clean period, and the background
period are presented in Table 2. Excluding cloudy and rainy days, the daytime peak of $J(O^1D)$ was
near $2.8 \times 10^5$ $s^{-1}$, indicating a high photochemical oxidation potential. As a secondary photochemical
product, $O_3$ exhibited a typical daily profile, peaking at 140.5 ppbv throughout the campaign. The
measurements of PNs peaked at 3.6 ppbv with a daytime average of $0.5 \pm 0.3$ ppbv, while ANs peaked
at 1.6 ppbv with a daytime average of $0.5 \pm 0.3$ ppbv. Ozone pollution periods were often associated
with high organic nitrates. The mean daily variation of $NO_x$ was consistent with the characteristics of
typical urban sites, significantly influenced by the morning-evening rush hours. During the daytime,
NO exhibited a single peak distribution, whereas $NO_2$ displayed a bimodal distribution. In comparison
to the background and clean period, the ozone pollution period was characterized with higher
temperatures and lower humidity. Additionally, the photolysis rate and levels of $PM_{2.5}$ were both
elevated during pollution days.

**Table 2.** Summary of daytime averages of chemical parameters over different periods during the Shanghai campaign.

| Pharse | Ozone pollution | Background | Clean |
| --- | --- | --- | --- |
| $T/°C$ | 29.8±3.7 | 27.0±3.4 | 26.0±3.5 |
| $P/hPa$ | 1043.6±0.8 | 1045.3±0.9 | 1044.3±1.4 |
| $RH/\%$ | 39.2±13.9 | 65.2±16.0 | 62.4±17.2 |
| $J(O^1D)\times10^5/s$ | 1.3±0.9 | 0.9±0.8 | 0.8±0.8 |
| $J(NO_2)\times10^3/s$ | 4.5±2.1 | 2.8±2.0 | 2.6±1.9 |
| $NO_2$/ppbv | 17.3±6.1 | 16.5±5.8 | 20.3±7.4 |
| $NO$/ppbv | 3.2±2.6 | 4.0±2.7 | 4.2±3.7 |
| $O_3$/ppbv | 78.6±30.9 | 41.6±27.7 | 45.0±21.5 |
| $PM_{2.5}/\mu g\cdot m^{-3}$ | 25.9±4.3 | 18.3±13.4 | 21.9±10.0 |
| $SO_2$/ppbv | 2.2±1.7 | 0.4±0.5 | 0.6±0.7 |
| $CO$/ppbv | 505.3±64.3 | 441.6±133.3 | 535.0±147.8 |
| $ISO$/ppbv | 0.1±0.1 | 0.2±0.2 | 0.1±0.1 |

The mean diurnal profiles of organic nitrates and related parameters observed during the campaign are shown in Fig. 3. During the ozone pollution period, $NO_x$ exhibited a peak concentration at 3:00 a.m., indicating the transport of a polluted air mass to the site. In comparison to the clean period, daytime $NO_x$ was lower during the ozone pollution period, particularly at noon when NO dropped to as low as 1.7 ppbv. Correspondingly, ANs during the ozone pollution period were generally high, but the daily variation was not significant. Therefore, the sources of ANs were more complex during the ozone pollution period, involving both transport contribution and local production, which aligns with the significantly increased background $O_3$. During the clean period, the daytime peak of $O_3$ was notably reduced and occurred later in the day. The fluctuations in $NO_x$ were more closely associated with morning and evening rush hours. The daytime peak of PNs decreased from 2.6 ppbv to 1.4 ppbv. In addition, the diurnal profile of ANs displayed a more pronounced peak at noon. During the background period, there was a further decline in the daytime peaks of $NO_x$ compared to the clean period. The diurnal profile of $O_3$ exhibited similar trends, but the duration of high $O_3$ was significantly shortened. The levels of both PNs and ANs exhibited a decline, approaching the background concentrations.

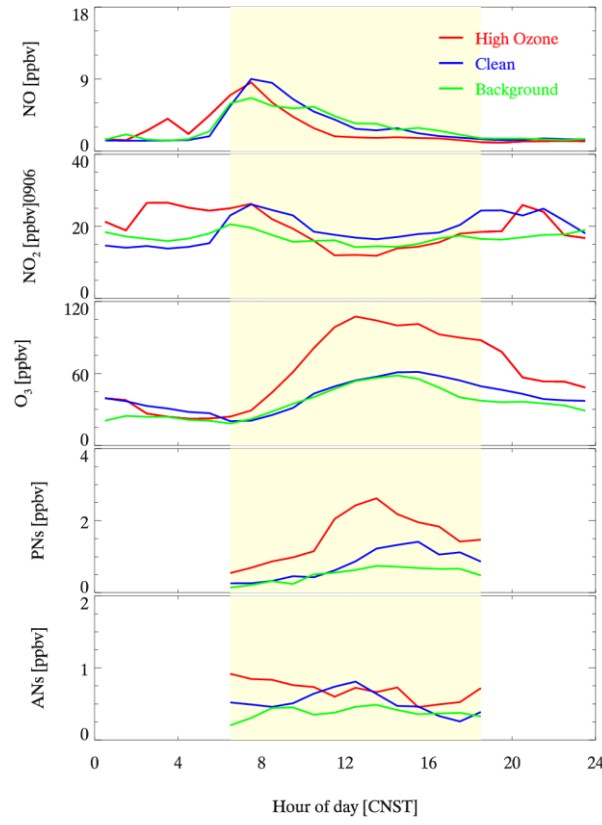

**Figure 3.** Mean diurnal profiles of organic nitrates and related parameters during different observation periods.

Here, we compare our observations with the study previously conducted in Xinjin, which is a suburban site, located in basin topography and faces emerging ozone pollution recently, to determine the effect of organic nitrate on $O_3$ production under different pollution conditions (Li et al., 2023). The Shanghai and Xinjin campaigns were conducted in early and late summer, respectively, exhibiting similar meteorological conditions. Photochemical conditions during both two campaigns are comparable, with the daily means of $J(O^1D)$ were $0.9 \times 10^{-5}$ $s^{-1}$ and $0.8 \times 10^{-5}$ $s^{-1}$, while the daily means of $J(NO_2)$ were $3.1 \times 10^{-3}$ $s^{-1}$ and $3.0 \times 10^{-3}$ $s^{-1}$, respectively, during Shanghai and Xinjin campaigns. The ratio of NO to $NO_2$ was 0.19 and 0.17 at Shanghai and Xinjin, respectively. Meanwhile, the concentration of $NO_x$ observed in Shanghai site (daily averages of 22.0 ppbv) is higher than that observed in Xinjin site (daily averages of 12.5 ppbv). The concentrations of $SO_2$ and CO at Shanghai site were 0.9 and 491.4 ppbv, while $SO_2$ and CO were 0.6 and 404.5 ppbv, respectively. Therefore, the air masses at Shanghai site were less aged than Xinjin site. However, the concentration of VOCs is lower in Shanghai campaign compared to Xinjin campaign, with daily mean of 23.5 ppbv compared to 22.4 ppbv. Therefore, a comparison of the two campaigns facilitates a comprehensive analysis of the impacts of organic nitrate chemistry on local ozone pollution.

## 3.2 Evaluation of organic nitrates simulations

In light of the updates to the mechanisms, validation testing has been conducted. Our previous study of the Xinjin campaign evaluated three mechanism schemes: mechanism S0, which is based on RACM2, mechanism S1 and mechanism S2 which refines the budget for BVOC-derived organic

nitrates (Li et al., 2023). It was found that the performance of mechanism S2 for organic nitrates exhibited an improvement exceeding 50%. Mechanism S2 has been updated by the Berkeley group (Fisher et al., 2016; Travis et al., 2016), which includes enhancements to the production mechanism of isoprene, the incorporation of the production mechanism for monoterpenes, and the completion of the uptake of organic nitrates by aerosols. Additionally, the Zare mechanism further refines the production mechanism of organic nitrates initiated by OH and $NO_3$, as well as improving the deposition process of organic nitrates. As a result, the Shanghai campaign was simulated using RACM2, Berkeley, and Zare mechanisms respectively for comparison.

The simulation result of organic nitrates under the three mechanisms is shown in Fig. 4a. The simulations for PAN or PNs exhibit an overall underestimation tendency, with the simulation of PAN demonstrating an even greater underestimation. Notably, the measured PNs remained above 500 pptv during nighttime, indicating a continuous transportation contribution at this site. Furthermore, the underestimation of PNs may be attributed to the unidentified $RO_x$ sources. It is consistent with the findings from summer campaigns in Wangdu, Beijing, where an underestimation of $RO_2$ was noted, particularly pronounced at elevated ambient $NO_x$ (Tan et al., 2017a). In terms of ANs, the simulation performances vary across different mechanisms. A significant overestimation of ANs is evident when utilized RACM2. Conversely, the simulation based on the Berkeley and Zare mechanisms generally results in an underestimation of ANs, while the underestimation of the Zare mechanism is more significant. Sensitivity tests conducted in Xinjin campaign suggested that the simple representation of ANs uptake caused the underestimation (Li et al., 2023), which is the same reason of underestimation in the Shanghai campaign. The uptake of ANs need further experimental data to achieve a detailed description to support the simulations.

The diurnal profile of simulated PNs is consistent with the measurements, both reaching their daytime peak shortly after sunrise. However, it is noteworthy that the peak concentration of PNs measurements is significantly higher than the simulation. In a similar pattern with PNs, the simulated ANs began to accumulate around 6:00 a.m. The measured ANs reached their peak near noon, whereas the simulations peaked at 3:00 pm. To evaluate the performance of simulations, as showed in Fig. 4b, three types of error ratios were calculated: Mean Square Error (MSE), Mean Absolute Error (MAE), and Mean Absolute Percentage Error (MAPE). Different error metrics for the organic nitrates exhibit a similar trend. The simulation performances of the Berkeley mechanism are better than the other two mechanisms. It should be noted that the Berkeley mechanism failed to fully reproduce the diurnal pattern of observed ANs. This is mainly due to the atmospheric transport that contributes to the ANs as mentioned in section 3.1. In addition, the drastic changes in $NO_x$ during rush hours will introduce errors to the ANs measurements. In addition, the Zare mechanism refined the oxidation of BVOCs by OH or $NO_3$ by introducing extra species with uncertain yields, which might bring biases to the simulations under high $NO_x$ and anthropogenic VOCs. In general, the Berkeley mechanism performs better on simulation of ANs than Zare mechanism. As a result, the subsequent analysis is based on the Berkeley mechanism.

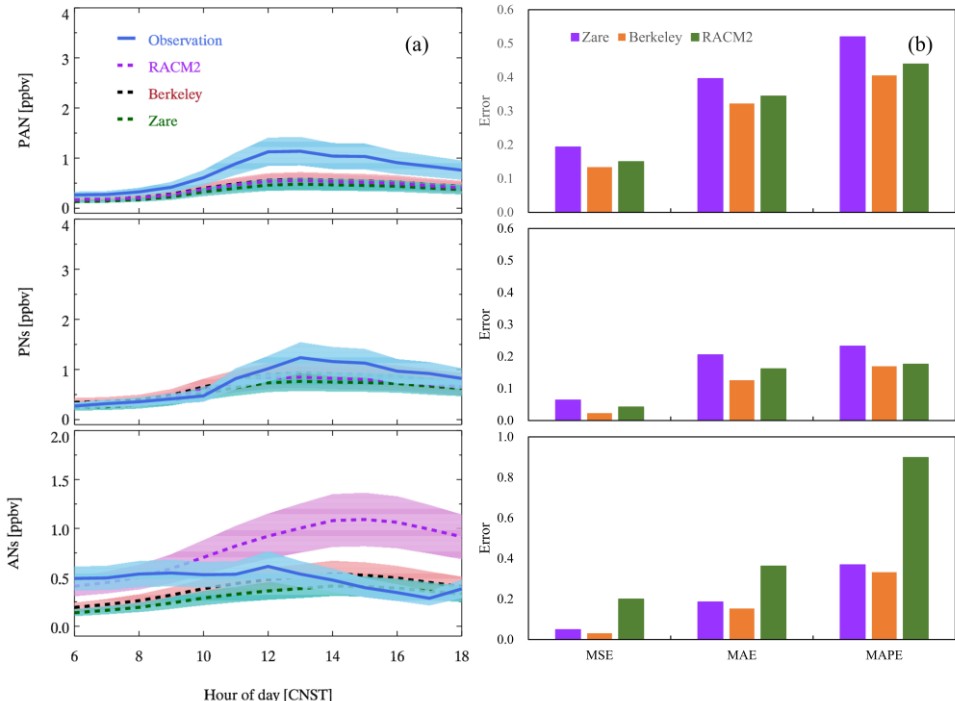

308

**Figure 4.** Mean diurnal profiles of observed and simulated ANs and PNs under different mechanism constraints during the Shanghai campaign (a), and the error of the different cases (b), including mean square error (MSE), mean absolute error (MAE) and mean absolute percentage error (MAPE).

## 3.3 Impact of PNs chemistry on local ozone production

Organic nitrates and $O_3$ have common precursors, and therefore the atmospheric behavior of organic nitrates has an important influence on the local $O_3$ distribution. The production of PNs consumes $NO_2$ and $RO_x$, thereby directly impacting $O_3$ production. The relationship between the distribution of PNs and $O_3$ is examined throughout the campaign. The observed PAN, PNs and $O_3$ between 9:00 a.m. and 2:00 p.m. are selected for the analysis to mitigate interference from sources that are not produced during daytime. The correlation of PAN or PNs with $O_3$ are shown in Fig. S2. Both PAN and PNs demonstrate a strong correlation with $O_3$ with the ratio of PAN or PNs to $O_3$ being 0.041 or 0.058. High ratios of PNs and $O_3$ usually indicate severe pollution episodes (Shepson et al., 1992; Sun et al., 2020; Zhang et al., 2023; Zhang et al., 2014). The minimum ratio of PNs to $O_3$ (0.024) was found during the clean periods, which can be regarded as the threshold for local photochemical pollution. $NO_x$ is the key pollutant for production of $O_3$ and PNs, in order to study the relationship between the ratio of PAN or PNs to $O_3$ and $NO_x$. The daytime ratios of PAN to $O_3$ derived from historical field observations are summarized with corresponding $NO_x$ concentrations in Fig. 5. The ratio derived from this study was distributed in the medium level of historical observations. The linear correlation of $NO_x$ and the ratio of PAN to $O_3$ ratio suggests that the $NO_x$ concentration controls the relative production of PNs and $O_3$.

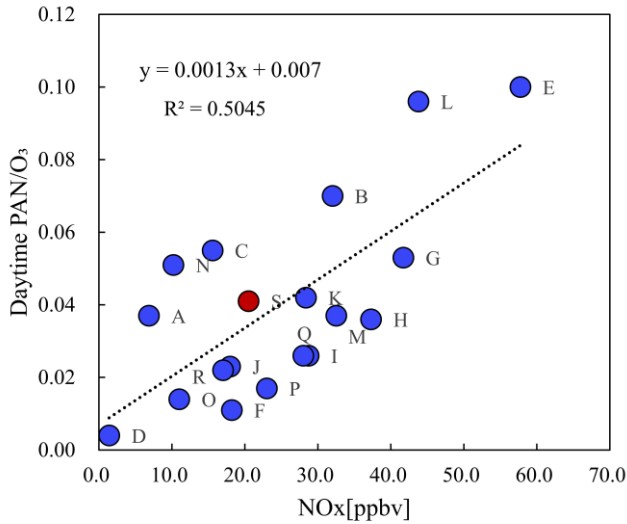

329

**Figure 5.** The relationship between historical daytime ratio of PAN to $O_3$ and $NO_x$ concentrations. The red dot is the Shanghai campaign, and the blue dots are the historical campaigns. A: Grosjean et al., 2002 (Grosjean et al., 2002); B: Lee et al., 2008 (Lee et al., 2008), C: Zhang et al., 2014 (Zhang et al., 2014), D-E: Zhang et al., 2009 (Zhang et al., 2009), F-G: Zeng et al., 2019 (Zeng et al., 2019), H-K: Zhang et al., 2019 (Zhang et al., 2019), L-M: Sun et al., 2020 (Sun et al., 2020); N: Li et al., 2023 (Li et al., 2023), O-R: Xu et al., 2024 (Xu et al., 2024), S: this study.

Sensitivity tests were conducted based on the box model to quantify the impact of PNs photochemistry on $O_3$ budgets. The differences of each pathway rate are calculated at the peak of $O_3$ production rate (Fig. 6). In the absence of PNs chemistry, two primary source pathways -namely, the reaction between $RO_2$ and NO, and the reaction between $HO_2$ and NO-exhibit large enhancements of 0.52 and 0.36 ppbv/h, respectively. In comparison, $O_3$ sinks increase slightly in the absence of PNs photochemistry, with the reaction between OH and $O_3$ showing the most significant enhancement of 0.11 ppbv/h. Therefore, during the Shanghai campaign, PNs photochemistry suppressed daytime ozone production mainly by reducing the reaction between $HO_2$ or $RO_2$ and NO.

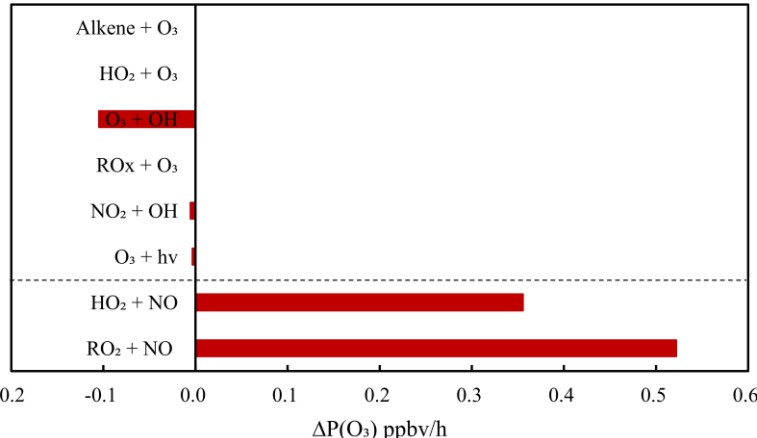


**Figure 6.** The simulated difference of ozone produce rate ($\Delta P(O_3)$) at 11am between the constraint of the PNs photochemistry and without the PNs photochemistry.

346       The PNs maintain a notable concentration until 6:00 p.m., suggesting a persistent impact on local

ozone production. As shown in Fig. 7a, the PNs photochemistry began to inhibit ozone production as
early as 6 a.m. and increased up to 0.8 ppbv/h (16%) at 10 a.m. The integrated inhibition of PNs
photochemistry on $O_3$ production was 4.5 ppbv in the Shanghai campaign (Fig. 7b), which was less
pronounced than the Xinjin campaign (20 ppbv). The reduced inhibition can be attributed to the lower
PNs production rate (P(PNs)) observed in the Shanghai campaign (Fig. S3), where the maximum
daytime P(PNs) was 0.89 ppbv/h, much lower than that in Xinjin campaign (3.09 ppbv/h). In addition,
the two campaigns had similar concentrations of VOCs, but daytime average of $NO_x$ in Shanghai site
is 22.0 ppbv, which is much higher than that of Xinjin site (10.2 ppbv). The PNs formation would be
reduced under high $NO_x$ condition due to the rapid termination reaction via OH and $NO_2$, and thus
limited the suppression effect of PNs formation which is the case in Shanghai campaign. Like in Xinjin
campaign, PAN chemistry suppressed $O_3$ formation at a rate of 2.84 ppbv/h at a suburban site in Hong
Kong (Zeng et al., 2019). However, it was reported that PAN tended to suppress $O_3$ production under
low-$NO_x$ and low-$RO_x$ conditions but enhanced $O_3$ production with sufficient $NO_x$ at a rural coastal
site in Qingdao, which is consistent with the comparison of Xinjin and Shanghai campaigns (Liu et al.,
2021). The impacts of PNs photochemistry on $O_3$ vary across different days. As shown in Fig. S4, the
integrated P($O_3$) change reaches 6.9 ppbv due to PNs photochemistry during ozone pollution period.
For the background and clean periods, the changes are close to each other with a value of 3.8 and 4.2
ppbv, respectively. Therefore, the PNs photochemistry contributes to more P($O_3$) inhibition during the
ozone pollution period, which should be considered in ozone pollution prevention.

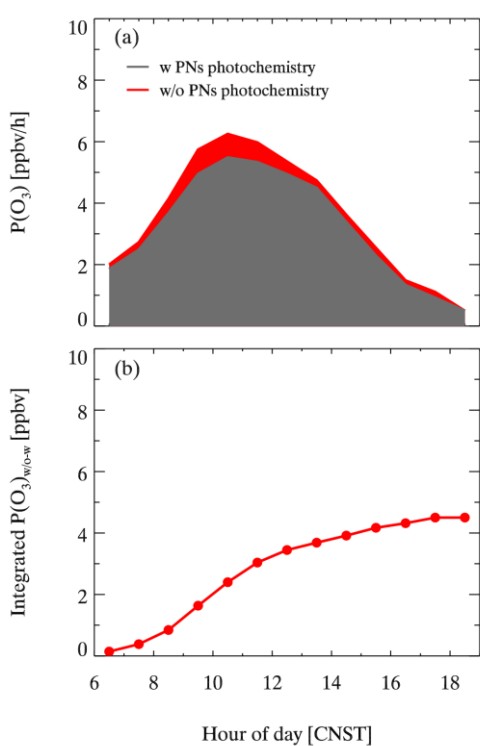


**Figure 7.** The impact of PNs photochemistry on P($O_3$) during the Shanghai campaign (a) daily changes of P($O_3$)
under the constraint of PNs photochemistry, (b) integrated P($O_3$) change constrained by PNs photochemistry.

## 3.4 Impact of ANs chemistry on local ozone production

To elucidate the impact of the α on $O_3$ production, the EKMA was utilized to investigate the combined response of $NO_x$ and VOCs to $O_3$ production at different α. The $O_3$ production was calculated by a simplified approach in method 2.2 and the α values were derived from weighted average of α based on the measured VOCs, the corresponding OH reaction rate constant and the α (Table S1) in Shanghai and Xinjin campaign, respectively. The model is initiated by the daytime averages of the environmental parameters. A comparative analysis is conducted between the Xinjin campaign and the Shanghai campaign where effective α is determined to be 0.031 and 0.053, respectively. As illustrated in Fig. 8a&b, $P(O_3)$ exhibits a similar trend with the variations of $NO_x$ and VOCs under different α, while the value of $P(O_3)$ reduces with larger α at the same levels of precursors. For example, when VOCs is at 8 ppbv and $NO_x$ reaches 9 ppbv, the $P(O_3)$ is 30.4 ppbv/h with α of 0.031, whereas it decreases to 24.6 ppbv/h when α is 0.053. In addition, the larger of α in the Shanghai campaign increases the threshold of $NO_x$ concentration for the transition of $O_3$ production regime. When the concentration of VOCs is fixed, a higher effective α results in a lower $NO_x$ concentration corresponding to the peak of $P(O_3)$. Consequently, an increase in α suppresses the peak of $P(O_3)$ and simultaneously affects its sensitivity to $NO_x$ and VOCs concentrations.

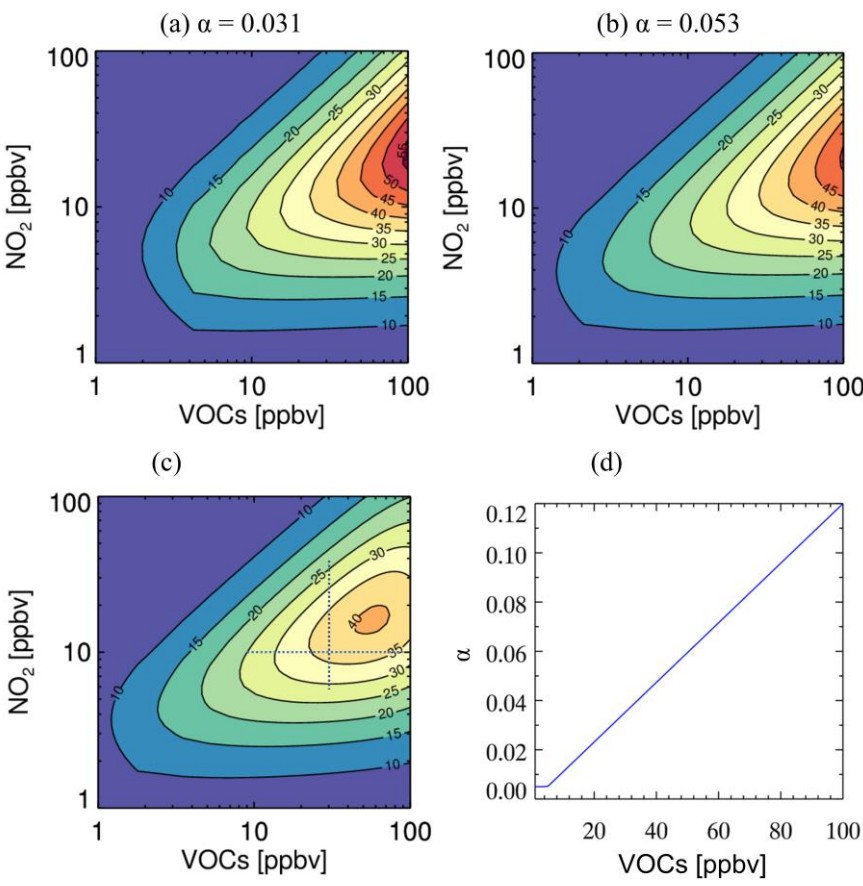

**Figure 8.** Ozone production ($P(O_3)$, ppb h-1) derived from a simplified analytic model is plotted as a function of $NO_x$ and VOCs under three different organic nitrate scenarios with branching ratios of (a) 0.031 for the Xinjin campaign, (b) 0.053 for the Shanghai campaign, and (c)VOC-dependent branching ratios for Shanghai campaign, where the

branching ratio decreases linearly from 12 to 0.5% with VOCs from 100 to 5 ppbv as shown in (d).
In the real atmosphere, the effective $\alpha$ of ANs tends to exhibit a decline with the reduction of
VOCs concentration. Historical studies show the general range from 0.03 to 0.04 in rural sites and
from 0.04 to 0.10 in urban environments, depending on the composition of VOCs and the $\alpha$ for BVOCs
(Farmer et al., 2011; Perring et al., 2010; Perring et al., 2013; Perring et al., 2009; Rosen et al., 2004b).
For simplicity, we use a linear relationship between $\alpha$ and VOC concentration in the sensitivity analysis,
as shown in Fig. 8d. An $\alpha$ value of 0.005 was selected for clean condition with VOC concentration less
than 5 ppbv, while 0.12 was selected for polluted condition with VOC concentration larger than 100
ppbv. The lower limit of 0.005 is the average of the $\alpha$ for methane and ethylene. The upper limit of
0.12 is set as the reported value of the $\alpha$ for isoprene and the $\alpha$ for aromatic hydrocarbons are generally
distributed around 0.1 (Perring et al., 2013). The assumption of this linear relationship between $\alpha$ and
VOC concentration has also been applied in a previous study (Farmer et al., 2011). With a varying $\alpha$,
as shown in Fig. 8d, $P(O_3)$ does not follow a consistent downward trend as VOCs decrease in VOC-
limited regime or transition regime. Instead, with the decrease of VOCs, the $P(O_3)$ is likely to increase
at first at a relatively high VOCs distribution, and then decrease similar to the fixed $\alpha$ scenario. Take
the cases of the horizontal dashed line as an example, at a fixed $NO_x$, the $P(O_3)$ increases as the VOCs
decrease within the range of about 60 to 100 ppbv, whereas $P(O_3)$ subsequently decrease as VOCs fell
below 60 ppbv. Therefore, with the reduction in VOCs emission, an increase in $\alpha$ directly correlates
with a reduction in the $P(O_3)$ peak. As a result, a positive correlation between $\alpha$ and VOCs
concentrations in real atmosphere might alter the $NO_x$-VOCs-$O_3$ relationship and diminish the effects
of VOCs reduction on ozone control.
Scenarios with different VOCs reactivity and $\alpha$ are selected for sensitivity tests to further
investigate the impact of ANs chemistry on the $O_3$ pollution control strategy in Shanghai. As illustrated
in Fig. 9a, variations of $P(O_3)$ among three scenarios exhibit an initial increase followed by a
subsequent decrease with rising $NO_x$. For the typical VOC reactivity and $\alpha$ obtained from the Shanghai
campaign, the turning point from $NO_x$ benefit to $NO_x$ limitation for $P(O_3)$ occurs at $NO_x$ concentration
of 1.38 ppbv, when $P(O_3)$ reaches a peak of 33.0 ppbv/h. When VOCs are reduced by 20% without
accounting for the reductions in $\alpha$, the turning point for $NO_x$ decreases to 1.26 ppbv with the $P(O_3)$
peak decreasing to 30.1 ppbv/h. When the reduction of $\alpha$ is considered alongside the decrease in VOCs
($\alpha$ decreases to 0.0265), the peak of $P(O_3)$ remains the same as the initial case. Consequently,
neglecting the $\alpha$ changes is likely to overestimate the effectiveness of emission control. Our
observations indicated that $NO_x$ in Shanghai was notably high, which accords with the conditions to
the right of the turning point in Fig. 9a. In this case, the major chain-termination reaction of the $HO_x$
cycle is the reaction between OH and $NO_2$ to produce $HNO_3$, while the share of the reaction that
produces ANs through the reaction between $RO_2$ and NO becomes relatively minor. As illustrated in
Fig. 9a, when $NO_x$ changes from 22.0 to 1.0 ppbv, the impact of $\alpha$ change will be larger, as the $P(O_3)$
difference between the two cases ranges from 0.1 to 2.6 ppbv/h. Therefore, the variation of $\alpha$ has a
limited impact on $O_3$ production at high $NO_x$, whereas it offsets the impact of VOCs reduction as $NO_x$
decrease to around 1.5 ppbv which represents a low-$NO_x$ emission condition. In addition, the
sensitivity analyses in a reduced VOC condition show that neglecting the $\alpha$ change still overestimates
the impact of VOCs reduction on $P(O_3)$ by around 4 times with $NO_x$ of 1 ppbv (Fig. 9b), which is also
more significant than the case in Shanghai campaign. Therefore, the variation in $\alpha$ has a temporarily
limited impact on $O_3$ production, whereas it should be seriously considered as $NO_x$ levels continue to
decrease.

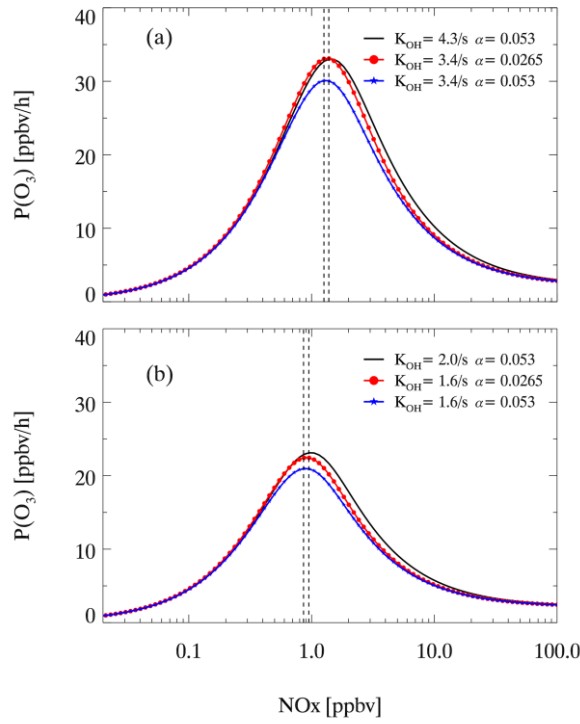

**Figure 9.** The ozone production rate (P(O₃)) varies as a function of $NO_x$ under different VOC-$NO_x$ regimes during Shanghai campaign: (a) under mean measured parameters during the whole campaign (solid line, VOC reactivity ($K_{OH}$) of 4.3/s, ANs branching ratio ($\alpha$) of 0.053); a 20% reduction in $K_{OH}$ with a 50% reduction in $\alpha$ (red dot line, 3.4/s, 0.0265); a 20% reduction in $K_{OH}$ with no change in $\alpha$ (blue dot line, 3.4/s, 0.053). (b) under observed parameters during the clean days (solid line, $K_{OH}$ of 2.0/s, $\alpha$ of 0.053); a 20% reduction in $K_{OH}$ with a 50% reduction in $\alpha$ (red dot line, 1.6/s, 0.0265); a 20% reduction in $K_{OH}$ with no change in $\alpha$ (blue dot line, 1.6/s, 0.053). Dash lines show the turning point in different cases.

To further investigate the effect of ANs formation on O₃ production during different days, sensitivity tests on VOCs reactivity and $\alpha$ are conducted based on typical conditions during different periods. The $\alpha$ values are derived as 0.055, 0.054 and 0.052, for the high ozone, clean and background periods, respectively. As shown in Fig. S4, the P(O₃) exhibits a similar trend with the increase of $NO_x$ across different periods. The P(O₃) peak during the background period (30.3 ppbv/h) is slightly lower than that during both the high ozone days and the clean days (32.5 and 32.4 ppbv/h). Therefore, the ANs chemistry has similar effects on O₃ production within different periods during the Shanghai campaign. Further comparisons of ozone production under varying precursor levels were conducted using historical observations collected in August 1994 at Mecklenburg-Vorpommern Mankmoos (MK), Germany (Ehhalt, 1999), and during the spring of 2006 in Mexico City (MX) (Farmer et al., 2011; Perring et al., 2010). The MK site serves as a typical clean background location with a very low effective $\alpha$ of 0.005, corresponding to $\tau$VOC of 0.4 s⁻¹, where methane is the predominant pollutant. Conversely, the MX site is characterized as an urban environment with an effective $\alpha$ of 0.036, where a total of 58 VOCs was measured, corresponding to $\tau$VOC of 3.1s⁻¹. The MK site shows a peak of P(O₃) is 2.2 ppbv/h at the $NO_x$ of 0.63 ppbv. In contrast, the MX site demonstrates a peak P(O₃) of 7.2 ppbv/h at a $NO_x$ of 1.9 ppbv. Given that the Xinjin and Shanghai sites exhibit higher VOCs reactivity

than MX, the corresponding peak $P(O_3)$ and the $NO_x$ inflection point are significantly elevated. This increase is primarily attributed to the high $P(HO_x)$, coupled with a low $\alpha$, which substantially enhances $P(O_3)$ under the intensified $HO_x$ cycling. Consequently, the ozone production potentials of urban sites in China are overall higher than in other regions, while the influence of $\alpha$ appears to be weak.

**4. Conclusions**

This study reveals the abundances of PNs and ANs and quantifies their respective impacts on O3 pollution based on the field campaign in Shanghai. They both showed higher values but less pronounced diurnal variation during the $O_3$ pollution period than the clean period. The mechanism validation indicates that Berkeley mechanism generally outperforms in the simulation of organic nitrates. The ratio of PNs/$O_3$ serves as a significant indicator of photochemistry. In comparison to the previous Xinjin campaign, the inhibition effect of PNs chemistry on daytime $O_3$ production diminished, likely attributed to the lower production of PNs. For ANs, the model simulation demonstrated that the branching ratio ($\alpha$) influences the $NO_x$-VOCs-$O_3$ sensitivity. The consideration of $\alpha$ value not only alters the $P(O_3)$ peak in EKMA but also resulted in low effectiveness of precursor reductions, as the $\alpha$ would change with the reduction of VOCs. It is worth mentioning that the complex polluted regions are usually characterized by high $NO_x$ and $HO_x$. In that case, the contribution of chain-termination reactions that produce ANs could be reduced, leading to limited impact of AN chemistry on $O_3$ formation. The effect of ANs chemistry on $O_3$ pollution control is therefore expected to enhance with further precursor reductions, and we suggest a pressing need for more measurements and analysis of organic nitrates to address the forthcoming challenges in air pollution mitigation.

**Code/Data availability.** The datasets used in this study are available from the corresponding author upon request (chenxr95@mail.sysu.edu.cn; k.lu@pku.edu.cn).

**Author contributions.** K.D.L. and X.R.C. designed the study. C.M.L. and X.R.C. analyzed the data and wrote the paper with input from K.D.L.

**Competing interests.** The authors declare that they have no conflicts of interest.

**Acknowledgments.** This work was supported by the National Natural Science Foundation of China (Grants No. 42407139); the National Natural Science Foundation of China (Grants No. 22406204); the special fund of State Environmental Protection Key Laboratory of Formation and Prevention of Urban Air Pollution Complex (SEPAir-2024080219); the Innovative Exploration Program of National Institute of Metrology, China (No. AKYCX2313).

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
