# Peer review of "The impact of organic nitrates on summer ozone formation in Shanghai,"

_EGUsphere, 2024_

## Referee Comment (RC1)

The study entitled "The Impact of Organic Nitrates on Summer Ozone Formation in Shanghai, China" by Li et al. investigates the role of alkyl nitrates (ANs) and peroxy nitrates (PNs) in ozone ($O_3$) formation using field measurements and modeling. The authors highlight that PNs suppress $O_3$ production, and the production yields ($\alpha$) of ANs significantly influence the sensitivity of $O_3$ formation to its precursors. These findings are critical for informing future air pollution mitigation policies. While the study is well-conducted and generally well-organized, certain areas require clarification or further analysis before publication.

Major comments:

1. The abstract mentions that "scenario analyses revealed that production yields ($\alpha$) of ANs would alter the response of $O_3$ formation to precursors due to varying compositions of volatile organic compounds." However, the sensitivity tests appear to use a linear relationship between $\alpha$ and VOC concentrations, as indicated in Fig. 8c. It is unclear how this linear relationship was derived. How does $\alpha$ vary with VOC composition in the sensitivity tests? Are there other factors, such as NOx levels, that might influence the variation in $\alpha$? Adding context in the introduction about the derivation of $\alpha$ and its dependencies would greatly improve clarity.

2. The findings on the localized impact of organic nitrates in Shanghai are valuable but would benefit from broader context. For instance, how does the suppression effect of PNs on $O_3$ in Shanghai compare to other urban or cleaner environments? What factors might explain differences in the impacts across various locations? Expanding the discussion to include comparisons with other field measurements would help strengthen the generalizability of the study's conclusions.

3. The authors emphasize that organic nitrate chemistry should inform future policy decisions. However, the study indicates that the impact of varying $\alpha$ on $O_3$ production in Shanghai is insignificant, likely associated with high NOx levels at present. To highlight the increasing importance of organic nitrates in future scenarios, I recommend conducting sensitivity analyses with reductions in NOx (and VOC) emissions. These additional simulations would demonstrate the evolving role of organic nitrates under cleaner air conditions and provide stronger policy-relevant insights.

Minor comments:

1. Line 54: The statement, "which are produced from $RO_2$ in the presence of oxidants...", is incorrect.
2. Section 2.2: It is unclear how the impact of PNs on $P(O_3)$ was quantified. Please provide a detailed explanation of the methodology.
3. Section 3.3: The expression "a/b" is ambiguous, as it could imply either "a or b" or "the ratio of a to b."
4. Figure 8: The caption does not include a description of panel (c). Clarify whether Fig. 8d represents simulations in Shanghai with VOC-dependent $\alpha$.

5. Line 388: The phrase "particularly in high NOx environments" appears contradictory to the statement on Line 391.

6. Line 389: The text refers to Fig. 10, but there is no corresponding figure in the main text.

7. Line 403: The term $\tau$VOC should be clearly defined the first time it is introduced.

---

## Author Comment (AC1)

**Response to Referee # *1***

All of the line numbers refer to Manuscript No.: EGUSPHERE-2024-3337.

Title: The Impact of Organic Nitrates on Summer Ozone Formation in Shanghai, China

Journal: Atmospheric Chemistry and Physics

We thank the reviewers' valuable comments and suggestions, we responded to the comments point to point, and revised the manuscript carefully. As detailed below, the referee's comments are shown in italicized font, our response is in orange, and new or modified text is in blue.

Reviewers/Editor comments:

**Referee #1:**

*The study entitled "The Impact of Organic Nitrates on Summer Ozone Formation in Shanghai, China" by Li et al. investigates the role of alkyl nitrates (ANs) and peroxy nitrates (PNs) in ozone ($O_3$) formation using field measurements and modeling. The authors highlight that PNs suppress $O_3$ production, and the production yields ($\alpha$) of ANs significantly influence the sensitivity of $O_3$ formation to its precursors. These findings are critical for informing future air pollution mitigation policies. While the study is well-conducted and generally well-organized, certain areas require clarification or further analysis before publication.*

Major comments:

1. *The abstract mentions that "scenario analyses revealed that production yields ($\alpha$) of ANs would alter the response of $O_3$ formation to precursors due to varying compositions of volatile organic compounds." However, the sensitivity tests appear to use a linear relationship between $\alpha$ and VOC concentrations, as indicated in Fig. 8c. It is unclear how this linear relationship was derived. How does $\alpha$ vary with VOC composition in the sensitivity tests? Are there other factors, such as NOx levels, that might influence the variation in $\alpha$? Adding context in the introduction about the derivation of $\alpha$ and its dependencies would greatly improve clarity.*

Thanks for your suggestion. The branching ratio ($\alpha$), the reaction ratio $k_b/(k_a+k_b)$, are different for various VOCs. Here, we use the effective $\alpha$ to analyze the effect of ANs chemistry on ozone production, which was calculated by weighted average based on the distribution of VOCs and the corresponding $\alpha$. Therefore, the change of VOCs composition would change the effective $\alpha$. As mentioned in our main text, the VOC compositions in polluted areas usually corresponds to high distribution of effective $\alpha$ and clean areas are linked to low distribution of effective $\alpha$ (Rosen et al., 2004;Perring et al., 2010;Perring et al., 2009;Perring et al., 2013;Farmer et al., 2011).Here, for simplicity, $\alpha$ are set to show a linear decrease with the decrease of VOCs to describe the relationship, which has also been used in the previous study (Farmer et al., 2011). To improve clarity, we have added the discussion as below "For simplicity, we use a

linear relationship between α and VOC concentration in the sensitivity analysis, as shown in Fig. 8d. An α value of 0.005 was selected for clean condition with VOC concentration less than 5 ppbv, while 0.12 was selected for polluted condition with VOC concentration larger than 100 ppbv. The lower limit of 0.005 is the average of the α for methane and ethylene. The upper limit of 0.12 is set as the reported value of the α for isoprene and the α for aromatic hydrocarbons are generally distributed around 0.1 (Perring et al., 2013). The assumption of this linear relationship between α and VOC concentration has also been applied in a previous study (Farmer et al., 2011)."

According to current research, there are limited evidences showing the potential influence of $NO_x$ level on α value. The values of α are associated with the carbon chain structure of the VOC molecule, the distribution of functional groups, temperature, and pressure (Reisen et al., 2005;Arey et al., 2001;Wennberg et al., 2018;Russell and Allen, 2005;Butkovskaya et al., 2012;Cassanelli et al., 2007), which varies between 0.1-35% (Table S1).

In addition, the definition of α and its dependencies are added in the introduction as "For ANs formation, the branching ratio (α), the reaction ratio $k_{1b}/(k_{1a}+k_{1b})$, varies between 0.1-35%, which are associated with the carbon chain structure of the molecule, the distribution of functional groups, temperature, and pressure (Reisen et al., 2005;Arey et al., 2001;Wennberg et al., 2018;Russell and Allen, 2005;Butkovskaya et al., 2012;Cassanelli et al., 2007)."

$$RO_2+NO \rightarrow RO+NO_2 \quad (R1a)$$
$$RO_2+NO \rightarrow RONO_2+NO_2 \quad (R1b)$$

2. *The findings on the localized impact of organic nitrates in Shanghai are valuable but would benefit from broader context. For instance, how does the suppression effect of PNs on $O_3$ in Shanghai compare to other urban or cleaner environments? What factors might explain differences in the impacts across various locations? Expanding the discussion to include comparisons with other field measurements would help strengthen the generalizability of the study's conclusions.*

Thanks for pointing out the issue. We have compared the impact of organic nitrates in SH to that in XJ, which is under a cleaner environment. To better present the generalizability of our study, we further compare our results with previous studies on the effect of PAN chemistry on $O_3$ production derived from other places. The revised texts and additional comparison are revised as follows. "Here, we compare our observations with the study previously conducted in Xinjin, which is a suburban site, located in basin topography and faces emerging ozone pollution recently, to determine the effect of organic nitrate on $O_3$ production under different pollution conditions (Li et al., 2023)." The comparison analysis about the effect of PNs on $O_3$ in section 3.3 is added as below. "The integrated inhibition of PNs photochemistry on $O_3$ production was 4.5 ppbv in the Shanghai campaign (Fig. 7b), which was less pronounced than the Xinjin campaign (20 ppbv). The reduced inhibition can be attributed to the lower PNs production rate (P(PNs)) observed in the Shanghai campaign (Fig. S3), where the maximum daytime P(PNs) was 0.89 ppbv/h, much lower than that in Xinjin campaign (3.09 ppbv/h). In addition, the two campaigns had similar concentrations of VOCs, but daytime average of $NO_x$ in Shanghai site is 22.0 ppbv, which is much higher than that of Xinjin site (10.2 ppbv). The PNs formation would be reduced under high $NO_x$ condition due to the rapid termination reaction via OH and $NO_2$, and thus limited the

suppression effect of PNs formation which is the case in Shanghai campaign. Like in Xinjin campaign, PAN chemistry suppressed $O_3$ formation at a rate of 2.84 ppbv/h at a suburban site in Hong Kong (Zeng et al., 2019). However, it was reported that PAN tended to suppress $O_3$ production under low-$NO_x$ and low-$RO_x$ conditions but enhanced $O_3$ production with sufficient $NO_x$ at a rural coastal site in Qingdao, which is consistent with the comparison of Xinjin and Shanghai campaigns (Liu et al., 2021)."

3. *The authors emphasize that organic nitrate chemistry should inform future policy decisions. However, the study indicates that the impact of varying α on $O_3$ production in Shanghai is insignificant, likely associated with high NOx levels at present. To highlight the increasing importance of organic nitrates in future scenarios, I recommend conducting sensitivity analyses with reductions in NOx (and VOC) emissions. These additional simulations would demonstrate the evolving role of organic nitrates under cleaner air conditions and provide stronger policy-relevant insights.*

Thanks for your suggestion. The related analysis has been provided in section 3.4. To improve the clarity, "Take the cases of the horizontal dashed line as an example, at a fixed $NO_x$, the $P(O_3)$ start to increase as the VOCs decrease from 100 to about 60 ppbv, and subsequently decrease as VOCs concentrations continue to decrease. Therefore, an increase in α directly correlates with a reduction in the $P(O_3)$ peak." is revised as "Take the cases of the horizontal dashed line as an example, at a fixed $NO_x$, the $P(O_3)$ increases as the VOCs decrease within the range of about 60 to 100 ppbv, whereas $P(O_3)$ subsequently decrease as VOCs fell below 60 ppbv. Therefore, with the reduction in VOCs emission, an increase in α directly correlates with a reduction in the $P(O_3)$ peak." Cases representing cleaner condition (low $NO_x$ and VOCs) is added in Fig. 8b as suggested by reviewer. The additional sensitivity analyses in section 3.4 are added as follows: "Scenarios with different VOCs reactivity and α are selected for sensitivity tests to further investigate the impact of ANs chemistry on the $O_3$ pollution control strategy in Shanghai. As illustrated in Fig. 9a, variations of $P(O_3)$ among three scenarios exhibit an initial increase followed by a subsequent decrease with rising $NO_x$. For the typical VOC reactivity and α obtained from the Shanghai campaign, the turning point from $NO_x$ benefit to $NO_x$ limitation for $P(O_3)$ occurs at $NO_x$ concentration of 1.38 ppbv, when $P(O_3)$ reaches a peak of 33.0 ppbv/h. When VOCs are reduced by 20% without accounting for the reductions in α, the turning point for $NO_x$ decreases to 1.26 ppbv with the $P(O_3)$ peak decreasing to 30.1 ppbv/h. When the reduction of α is considered alongside the decrease in VOCs (α decreases to 0.0265), the peak of $P(O_3)$ remains the same as the initial case. Consequently, neglecting the α changes is likely to overestimate the effectiveness of emission control. Our observations indicated that $NO_x$ in Shanghai was notably high, which accords with the conditions to the right of the turning point in Fig. 9a. In this case, the major chain-termination reaction of the $HO_x$ cycle is the reaction between OH and $NO_2$ to produce $HNO_3$, while the share of the reaction that produces ANs through the reaction between $RO_2$ and NO becomes relatively minor. As illustrated in Fig. 9a, when $NO_x$ changes from 22.0 to 1.0 ppbv, the impact of α change will be larger, as the $P(O_3)$ difference between the two cases ranges from 0.1 to 2.6

ppbv/h. Therefore, the variation of α has a limited impact on O₃ production at high NOₓ, whereas it offsets the impact of VOCs reduction as NOₓ decrease to around 1.5 ppbv which represents a low-NOₓ emission condition. In addition, the sensitivity analyses in a reduced VOC condition show that neglecting the α change still overestimates the impact of VOCs reduction on P(O₃) by around 4 times with NOₓ of 1 ppbv (Fig. 9b), which is also more significant than the case in Shanghai campaign."

[Figure]

**Figure 9.** The ozone production rate (P(O₃)) varies as a function of NOₓ under different VOC-NOₓ regimes during Shanghai campaign: (a) under mean measured parameters during the whole campaign (solid line, VOC reactivity (K$_{OH}$) of 4.3/s, ANs branching ratio (α) of 0.053); a 20% reduction in K$_{OH}$ with a 50% reduction in α (red dot line, 3.4/s, 0.0265); a 20% reduction in K$_{OH}$ with no change in α (blue dot line, 3.4/s, 0.053). (b) under observed parameters during the clean days (solid line, K$_{OH}$ of 2.0/s, α of 0.053); a 20% reduction in K$_{OH}$ with a 50% reduction in α (red dot line, 1.6/s, 0.0265); a 20% reduction in K$_{OH}$ with no change in α (blue dot line, 1.6/s, 0.053). Dash lines show the turning point in different cases.

Minor comments:
1. *Line 54: The statement, "which are produced from RO₂ in the presence of oxidants...", is incorrect.*

Thank you for pointing out the mistake. "which are produced from RO₂ in the presence of oxidants such as OH." is revised as "which produce RO₂ in the presence of oxidants, such as OH.".

2. *Section 2.2: It is unclear how the impact of PNs on P(O₃) was quantified. Please provide a detailed explanation of the methodology.*

Thank you for the suggestion. The methodology related with the impact of PNs on P(O₃) is added in section 2.2 as below "The impact of PNs photochemistry on local ozone is quantified by comparing the difference of the daytime P(O₃) between the scenarios with and without PNs photochemistry via a chemical box model. Here, the PNs photochemistry includes the production and removal of PAN, MPAN and PPN."

3. *Section 3.3: The expression "a/b" is ambiguous, as it could imply either "a or b" or "the ratio of a to b."*
The expression "PAN/PNs" and "PAN/O₃" have been replaced by "PAN or PNs" and "the ratio of PAN to O₃" accordingly.

4. *Figure 8: The caption does not include a description of panel (c). Clarify whether Fig. 8d represents simulations in Shanghai with VOC-dependent α.*
Thanks for pointing the issue. The Figure 8 is revised as below.

[Figure]

**Figure 8.** Ozone production (P(O₃), ppb h-1) derived from a simplified analytic model is plotted as a function of NOₓ and VOCs under three different organic nitrate scenarios with branching ratios of (a) 0.031 for the Xinjin campaign, (b) 0.053 for the Shanghai campaign, and (c) VOC-dependent branching ratios for Shanghai campaign, where the branching ratio decreases linearly from 12 to 0.5% with VOCs from 100 to 5 ppbv as shown in (d).

**ferences**

Arey, J., Aschmann, S. M., Kwok, E. S. C., and Atkinson, R.: Alkyl nitrate, hydroxyalkyl nitrate, and hydroxycarbonyl formation from the NOx-air photooxidations of C-5-C-8 n-alkanes, J. Phys. Chem. A, 105, 1020-1027, 10.1021/jp003292z, 2001.

Butkovskaya, N., Kukui, A., and Le Bras, G.: Pressure and Temperature Dependence of Methyl Nitrate Formation in the CH3O2+NO Reaction, J. Phys. Chem. A, 116, 5972-5980, 2012.

Cassanelli, P., Fox, D. J., and Cox, R. A.: Temperature dependence of pentyl nitrate formation from the reaction of pentyl peroxy radicals with NO, Phys. Chem. Chem. Phys., 9, 4332-4337, 10.1039/b700285h, 2007.

Farmer, D. K., Perring, A. E., Wooldridge, P. J., Blake, D. R., Baker, A., Meinardi, S., Huey, L. G., Tanner, D., Vargas, O., and Cohen, R. C.: Impact of organic nitrates on urban ozone production, Atmos. Chem. Phys., 11, 4085-4094, 10.5194/acp-11-4085-2011, 2011.

Liu, Y., Shen, H., Mu, J., Li, H., Chen, T., Yang, J., Jiang, Y., Zhu, Y., Meng, H., Dong, C., Wang, W., and Xue, L.: Formation of peroxyacetyl nitrate (PAN) and its impact on ozone production in the coastal atmosphere of Qingdao, North China, Sci. Total Environ., 778, 10.1016/j.scitotenv.2021.146265, 2021.

Perring, A. E., Bertram, T. H., Wooldridge, P. J., Fried, A., Heikes, B. G., Dibb, J., Crounse, J. D., Wennberg, P. O., Blake, N. J., Blake, D. R., Brune, W. H., Singh, H. B., and Cohen, R. C.: Airborne observations of total RONO2: new constraints on the yield and lifetime of isoprene nitrates, Atmos. Chem. Phys., 9, 1451-1463, 10.5194/acp-9-1451-2009, 2009.

Perring, A. E., Bertram, T. H., Farmer, D. K., Wooldridge, P. J., Dibb, J., Blake, N. J., Blake, D. R., Singh, H. B., Fuelberg, H., Diskin, G., Sachse, G., and Cohen, R. C.: The production and persistence of Sigma RONO2 in the Mexico City plume, Atmos. Chem. Phys., 10, 7215-7229, 2010.

Perring, A. E., Pusede, S. E., and Cohen, R. C.: An Observational Perspective on the Atmospheric Impacts of Alkyl and Multifunctional Nitrates on Ozone and Secondary Organic Aerosol, Chemical Reviews, 113, 5848-5870, 10.1021/cr300520x, 2013.

Reisen, F., Aschmann, S. M., Atkinson, R., and Arey, J.: 1,4-hydroxycarbonyl products of the OH radical initiated reactions of C-5-C-8 n-alkanes in the presence of N0, Environ. Sci. Technol., 39, 4447-4453, 10.1021/es0483589, 2005.

Rosen, R. S., Wood, E. C., Wooldridge, P. J., Thornton, J. A., Day, D. A., Kuster, W., Williams, E. J., Jobson, B. T., and Cohen, R. C.: Observations of total alkyl nitrates during Texas Air Quality Study 2000: Implications for O-3 and alkyl nitrate photochemistry, J. Geophys. Res.-Atmos., 109, 10.1029/2003jd004227, 2004.

Russell, M., and Allen, D. T.: Predicting secondary organic aerosol formation rates in southeast Texas, J. Geophys. Res.-Atmos., 110, 10.1029/2004jd004722, 2005.

Wennberg, P. O., Bates, K. H., Crounse, J. D., Dodson, L. G., McVay, R. C., Mertens, L. A., Nguyen, T. B., Praske, E., Schwantes, R. H., Smarte, M. D., St Clair, J. M., Teng, A. P., Zhang, X., and Seinfeld, J. H.: Gas-Phase Reactions of Isoprene and Its Major Oxidation Products, Chemical Reviews, 118, 3337-3390, 10.1021/acs.chemrev.7b00439, 2018.

Zeng, L., Fan, G.-J., Lyu, X., Guo, H., Wang, J.-L., and Yao, D.: Atmospheric fate of peroxyacetyl nitrate in suburban Hong Kong and its impact on local ozone pollution, Environ. Pollut., 252, 1910-1919, 10.1016/j.envpol.2019.06.004, 2019.

---

## Author Comment (AC2)

**Response to Referee # *2***

All of the line numbers refer to Manuscript No.: EGUSPHERE-2024-3337.

Title: The Impact of Organic Nitrates on Summer Ozone Formation in Shanghai, China

Journal: Atmospheric Chemistry and Physics

We thank the reviewers' valuable comments and suggestions, we responded to the comments point to point, and revised the manuscript carefully. As detailed below, the referee's comments are shown in italicized font, our response is in orange, and new or modified text is in blue.

Reviewers/Editor comments:

**Referee #2:**

*This study conducted field measurements of gaseous organic nitrates at an urban site and assessed their effects on local O3 production using a box model incorporating an updated mechanism. Organic nitrates formation and its contribution to SOA and effect on O3 are of great concern to the researchers.*

Major comments:

*My main question about this work is the performance of the model simulations. The author used three mechanisms to simulate ONs, but as can be seen in Figure 4, the model simulated the concentration levels of ANs but poorly simulated the trend of ANs. It appears that the trend of simulated and observed ANs are opposite. The effectiveness of the model simulation affects the later analysis of ANs and the authors need to give an explanation. What about the R2 of the model simulation? Why does Zare's model that considers the ON formation initiated by OH and NO3 not work well compared to other models? Some specific comments are also provided below.*

Thanks for the valuable comments. The simulated and measured levels of ANs are comparable, but there are some differences in the diurnal profiles, which may be caused by the following reasons. Firstly, the box model majorly simulates local chemistry budgets of ANs. As mentioned in 3.1 section, the transport might contribute to extra ANs from upwind areas. For example, on May 28th, there were high concentrations of both PNs and ANs in the morning accompanied by a change in wind direction and an increase in wind speed. In addition, the underestimation of PAN through box models has been reported in previous studies (Zeng et al., 2019;Sun et al., 2020;Xu et al., 2024). Secondly, the extremely high $NO_x$ in the morning and evening rush hours will introduce larger errors to the ANs measurement during these periods. In addition, the error ratios (MSE) between measured and simulated ANs concentration based on Zare and Berkeley mechanism are 0.049 and 0.031 respectively, which are capable of reproducing the ANs chemistry. The impact of ANs on $O_3$ production are mainly qualified during the daytime period with high $O_3$ production, when ANs are better modeled.

     The poor performance of the Zare mechanism on ON formation simulation can be

attributed to its modification of BVOCs oxidation by OH or NO$_3$ to produce ANs based on the Berkeley mechanism. The Zare mechanism refined the subsequent reactions of RO$_2$ generated by oxidation of BVOCs and would lead to a decrease in the ANs production from BVOCs. For example, after NO$_3$+ISO produce nitrooxy peroxy radical (INO$_2$), the Zare mechanism sets the C5 carbonyl nitrate (ICN) yield at 54% and 72% for INO$_2$+NO$_3$ and INO$_2$+INO$_2$, respectively. These modifications might introduce extra uncertainty since the yield of these newly added species could change in different conditions (e.g. high NO$_x$ and anthropogenic VOCs). To clarify this point, the following explanation is added in 3.2 section "It should be noted that the Berkeley mechanism failed to fully reproduce the diurnal pattern of observed ANs. This is mainly due to the atmospheric transport that contributes to the ANs as mentioned in section 3.1. In addition, the drastic changes in NO$_x$ during rush hours will introduce errors to the ANs measurements. In addition, the Zare mechanism refined the oxidation of BVOCs by OH or NO$_3$ by introducing extra species with uncertain yields, which might bring biases to the simulations under high NO$_x$ and anthropogenic VOCs. In general, the Berkeley mechanism performs better on simulation of ANs than Zare mechanism.".

Minor comments:

1.  *section 3.1, The authors divided the entire campaign into ozone pollution days, clean days, and background days. What are the specific criteria for dividing the clean and background days? How do precursors and environmental factors affect ANs and PNs on different days? The authors are also suggested to compare the effects of ANs and PNs on O3 on different days in the later analysis.*

Thanks for pointing the issue. To better clarify this point, we revised the description of classification in mentioned in 3.1 section as "The days without ozone pollution are categorized as clean or background days. For clean days, parameters, including K$_{OH}$, SO$_2$, and CO, show significant diurnal variations (Fig S1), and no rain occurs. The days that are neither ozone pollution days nor clean days are then classified as background days."

[Figure]

**Figure S1.** Mean diurnal profiles of VOC reactivity ($K_{OH}$), $SO_2$ and CO during different observation periods.

The factors affecting ANs and PNs are indeed an important topic, however, it is beyond scope of this study. We prefer to address this in our next study, where we will comprehensively analyze the budgets of ANs and PNs from a collections of several historical campaign case by case.

As suggested by reviewer, we further compare the effects of PNs and ANs on $O_3$ during different days. The results of comparison for PNs are added in section 3.3 and Fig. S3 as "The impacts of PNs photochemistry on $O_3$ vary across different days. As shown in Fig. S4, the integrated $P(O_3)$ change reaches 6.9 ppbv due to PNs photochemistry during ozone pollution period. For the background and clean periods, the changes are close to each other with a value of 3.8 and 4.2 ppbv, respectively. Therefore, the PNs photochemistry contributes to more $P(O_3)$ inhibition during the ozone pollution period, which should be considered in ozone pollution prevention.".

[Figure]

**Figure S4.** The integrated P(O₃) changes constrained by PNs photochemistry during different observation periods.

For ANs, the effects are simulated via simplified box model, due to the wide variety of ANs and the related complex mechanism. During the different days, the branching ratio of ANs are 0.055, 0.054 and 0.052, respectively. Therefore, the ANs chemistry are similar across different days. The effects of ANs formation on O₃ production during different days are added in section 3.4 as below.

"To further investigate the effect of ANs formation on O₃ production during different days, sensitivity tests on VOCs reactivity and $\alpha$ are conducted based on typical conditions during different periods. The $\alpha$ values are derived as 0.055, 0.054 and 0.052, for the high ozone, clean and background periods, respectively. As shown in Fig. S4, the P(O₃) exhibits a similar trend with the increase of NOₓ across different periods. The P(O₃) peak during the background period (30.3 ppbv/h) is slightly lower than that during both the high ozone days and the clean days (32.5 and 32.4 ppbv/h). Therefore, the ANs chemistry has similar effects on O₃ production within different periods during the Shanghai campaign."

2. *Line 242-256, compare ANs and PNs in Shanghai and Xinjin. And Line 329-337, the reduced inhibition of PNs on ozone production in Shanghai compared to Xinjin was attributed to the lower PNs production rate. The authors did not do further modeling analysis here, but it is suggested that the authors provide a reasonable discussion.*

Thanks for your suggestion. Detailed discussion is added in section 3.3 as "The integrated inhibition of PNs photochemistry on O₃ production was 4.5 ppbv in the Shanghai campaign (Fig. 7b), which was less pronounced than the Xinjin campaign (20 ppbv). The reduced inhibition can be attributed to the lower PNs production rate (P(PNs)) observed in the Shanghai campaign (Fig. S3), where the maximum daytime P(PNs) was 0.89 ppbv/h, much lower than that in Xinjin campaign (3.09 ppbv/h). In addition, the two campaigns had similar concentrations of VOCs, but daytime average of NOₓ in Shanghai site is 22.0 ppbv, which is much higher than that of Xinjin site (10.2 ppbv). The PNs formation would be reduced under high NOₓ condition due to the rapid

termination reaction via OH and $NO_2$, and thus limited the suppression effect of PNs formation which is the case in Shanghai campaign. Like in Xinjin campaign, PAN chemistry suppressed $O_3$ formation at a rate of 2.84 ppbv/h at a suburban site in Hong Kong (Zeng et al., 2019). However, it was reported that PAN tended to suppress $O_3$ production under low-$NO_x$ and low-$RO_x$ conditions but enhanced $O_3$ production with sufficient $NO_x$ at a rural coastal site in Qingdao, which is consistent with the comparison of Xinjin and Shanghai campaigns (Liu et al., 2021)."

3. *Line 378-411, The Fig. 10 was missing. Regarding the sensitivity of VOCs reactivity and α to the effect of ANs formation on O3 production, it is suggested that the authors make more comparisons of the results of the most recent studies or those from China. In addition, I would like to know if the authors have tested the sensitivity of VOCs reactivity and α for different days, i.e. ozone pollution, clean and background days and what are the differences in the results?*

Thank you for pointing out the mistake. The Fig. 10 should be Fig. 9. As the measurements of total ANs and PNs are pretty scarce in China, a comparison among different campaigns can be done in the future.

According to the suggestions by reviewer, we further test the sensitivity of $O_3$ production to VOCs reactivity and α for different periods. The results of sensitivity analysis are added as Fig. S4 and the detailed discussion are shown as below "To further investigate the effect of ANs formation on $O_3$ production during different days, sensitivity tests on VOCs reactivity and $α$ are conducted based on typical conditions during different periods. The $α$ values are derived as 0.055, 0.054 and 0.052, for the high ozone, clean and background periods, respectively. As shown in Fig. S5, the $P(O_3)$ exhibits a similar trend with the increase of $NO_x$ across different periods. The $P(O_3)$ peak during the background period (30.3 ppbv/h) is slightly lower than that during both the high ozone days and the clean days (32.5 and 32.4 ppbv/h). Therefore, the ANs chemistry has similar effects on $O_3$ production within different periods during the Shanghai campaign."

[Figure]

**Figure S5.** The ozone production rate ($P(O_3)$) varies as a function of $NO_x$ under different VOC-$NO_x$ regimes during Shanghai campaign: (a) under measured parameters during high ozone period; (b) during the clean period; (c) during the background period. The solid line shows the mean $K_{OH}$ with effective $\alpha$; the red dot line shows a 20% reduction in $K_{OH}$ with a 50% reduction in $\alpha$; the blue dot line shows a 20% reduction in $K_{OH}$ with no change in $\alpha$. Dash lines show the turning point in different cases.

**References**

Liu, Y., Shen, H., Mu, J., Li, H., Chen, T., Yang, J., Jiang, Y., Zhu, Y., Meng, H., Dong, C., Wang, W., and Xue, L.: Formation of peroxyacetyl nitrate (PAN) and its impact on ozone production in the coastal atmosphere of Qingdao, North China, Sci. Total Environ., 778, 10.1016/j.scitotenv.2021.146265, 2021.

Sun, M., Cui, J. n., Zhao, X., and Zhang, J.: Impacts of precursors on peroxyacetyl nitrate (PAN) and relative formation of PAN to ozone in a southwestern megacity of China, Atmos. Environ., 231, 10.1016/j.atmosenv.2020.117542, 2020.

Xu, T., Nie, W., Xu, Z., Yan, C., Liu, Y., Zha, Q., Wang, R., Li, Y., Wang, L., Ge, D., Chen, L., Qi, X., Chi, X., and Ding, A.: Investigation on the budget of peroxyacetyl nitrate (PAN) in the Yangtze River Delta: Unravelling local photochemistry and regional impact, Sci. Total Environ., 917, 10.1016/j.scitotenv.2024.170373, 2024.

Zeng, L., Fan, G.-J., Lyu, X., Guo, H., Wang, J.-L., and Yao, D.: Atmospheric fate of peroxyacetyl nitrate in suburban Hong Kong and its impact on local ozone pollution, Environ. Pollut., 252, 1910–1919, 10.1016/j.envpol.2019.06.004, 2019.